# Multidimensional health patterns and labor market participation among older workers: Evidence from a European six-year follow-up study

Isabel Baumann[1]*, Vanessa Gut[1,2], Rainer Gabriel[3], Daniel Fredriksson[4], Johan Fritzell[5]

**1** Institute of Public Health, ZHAW Zurich University of Applied Sciences, Winterthur, Switzerland, **2** Faculty of Health Sciences and Medicine, University of Lucerne, Lucerne, Switzerland, **3** School of Social Work, ZHAW Zurich University of Applied Sciences, Zürich, Switzerland, **4** Department of Sociology, Uppsala University, Uppsala, Sweden, **5** Aging Research Center, Karolinska Institutet and Stockholm University, Sweden

\* isabel.baumann@zhaw.ch

## Abstract

Health is a multidimensional concept and consists of different dimensions such as mental and physical health. In this study, we built on this conception of health by investigating the effects of multidimensional health patterns (MHP) on labor force participation among older workers. We used data from 5`006 older workers (mean age: 55.8 years; 56% female) from the Survey of Health, Ageing and Retirement in Europe. In a first step, we conducted a latent transition analysis with four health indicators (chronic diseases, limitations with activities, self-perceived health, and depressive symptoms) to identify the MHP. In a second step, we assessed the association of the MHP with workers' labor force participation six years after the first point of observation using Wald tests. Our analysis provided us with four MHP: (1) overall healthy workers, (2) workers with moderate activity limitations and low self-perceived health, (3) physically healthy workers with low self-perceived health and depressive symptoms, and (4) workers with overall poor health. We found differences between the MHP in job situation (p < 0.001) and changes in working hours (p < 0.001) four years later, with MHP 4 showing the highest probability of being permanently sick or disabled and MPH 2 and 3 showing the greatest reductions in working hours. Our results imply that physical health is a better predictor of labor force participation among older workers than other dimensions of health. Using health as a multidimensional construct thus allows to better understand the social implications of poor health.

**Data availability statement:** The data analyzed are free and publicly available after registration at the SHARE Research Data Center: https://releases.sharedataportal.eu/login?redirect=%-2Fusers%2Flogin This study used data from SHARE waves 5, 6, and 8 (DOIs: 10.6103/SHARE.w5.800, 10.6103/SHARE.w6.800, 10.6103/SHARE.w8.800). See Börsch-Supan et al. (2013) for methodological details.

**Funding:** This work was supported by the Swiss National Science Foundation (SNSF) (grant number 179696: IB, VG; https://data.snf.ch/grants/grant/179696); the ZHAW Schwerpunkt Angewandte Gerontologie [Research focus "Applied Gerontology": RG; https://www.zhaw.ch/de/forschung/departementsuebergreifende-kooperationen/angewandte-gerontologie; Forte (grant numbers 2016-07206: JF; and 2021-00999: DF; https://forte.se/en/about-forte/our-mission/). The funders had no role in study design, data collection and analysis, decision to publish, or preparation of the manuscript.

**Competing interests:** The authors have declared that no competing interests exist.

## 1. Introduction

In the past three decades, all member states of the European Union have reformed their pension systems to address demographic pressure caused by the aging of the population. In most cases, adaptations have been made by raising the retirement age. Other approaches have been the promotion of labor market participation among older workers through incentives or by allowing a higher degree of flexibility in the retirement process [1,2]. As a result, a trend toward a prolongation of working life has been observed [3].

This trend is reflected in the fact that the share of older workers holding paid employment is increasing. In countries of the Organisation for Economic Co-operation and Development (OECD), the employment rate of people aged 55–64 has increased from approximately 54% in 2007 (the earliest date for which data are available) to 64% in 2023 (the latest date for which data are available) [4]. Additionally, older workers now experience more varied retirement transitions [5] but generally remain engaged in paid employment until increasingly older ages, as suggested by the rise in the *observed* retirement age (i.e., the age at which people report exiting the labor market, independent of the *statutory* retirement age). According to self-reported information on retirement age, this age was 62.3 years for men and 61.0 years for women across the OECD countries in 2007. It increased by over two years to 64.4 years for men and 63.1 years for women as of 2022 [6]. In addition to the aforementioned policy changes, ongoing improvements in health and increases in educational levels also drive the labor market participation of older workers [7,8].

Despite the generally high level of labor force participation of older workers, a non-negligible share of these workers suffer from adverse health conditions at the end of their careers [9]. Some exit the labor force due to these adverse health conditions [10,11], while others remain [12]. Yet, other studies have indicated that various indicators of poor health have become less predictive for retirement than previously [13], which suggests that older workers remain in the labor market despite experiencing health problems. These findings lead us to our main question: which combinations of health factors, both mental and physical, play a role in determining whether older workers remain or exit the labor force.

Previous research has shown a variety of potential circumstances: Some older workers with adverse health conditions might be required to continue working involuntarily to secure their income or maintain their standard of living [14,15]. Other workers continue to work voluntarily; for example, they may choose to remain in the labor force because their working conditions are designed in such a way that they can continue to work despite the restrictions of health conditions [12,16]. More generally, a broad body of literature on work ability suggests that the likelihood of remaining in employment depends on whether a balance between an individual's resources (e.g., health, money or education) and the requirements for successfully continuing to work can be found [16–18]. However, a common limitation of the existing literature lies in the fact that it typically defines health by a single outcome. Much less is known about the relevance of different dimensions of health and the combination of these dimensions to the labor force participation of older workers. Recent research that includes

both poor mental and physical health as risk factors for early exit from the labor force has shown that we cannot infer from the results for one health indicator to a similar result for another indicator [19].

Therefore, in this study, we conceptualized health as a multidimensional construct—a definition that is in line with the World Health Organization (WHO)'s definition of health [12,20]. Following this definition, we assessed the role that multidimensional health plays in labor market participation. This approach consisted not merely of including multiple health indicators in conventional statistical models but began by identifying distinct subgroups characterized by qualitatively different *latent* multidimensional health patterns (MHP) [21–23]. These were based on the comprehensive model by Boehme and colleagues that included four primary indicators: chronic illness, functional limitation, self-perceived health, and depressive symptoms [24]. The advantage of this model is that it includes two indicators for mental health (i.e., depressive symptoms and self-perceived health) and it combines health dimensions that have individually been shown to be relevant to labor force participation among older workers. Notably, Hasselhorn and colleagues [12] pointed out that the association of poor health with an early exit from the labor force, which has been supported by many studies, is an average effect. Such average effects bear the risk of masking distinct and deviating mechanisms within subgroups (Hasselhorn et al., 2022). By identifying MHP that can be, e.g., purely related to mental health, physical health, or a combination of both, we provide a more comprehensive understanding of the heterogeneous mechanisms by which health is related to labor force participation among older workers.

On the one hand, we examined the stability of these MHP over a period of two years. On the other hand, we investigate how these MHP are associated with labor force participation among older workers six years later. As a result of its longitudinal design, our study provides a clear understanding of how different dimensions of health affected workers' employment trajectories. We thus contribute to the segment of life course research that investigates late careers and the transition to retirement.

## 2. The multidimensional nature of health among older workers

Most of the previous studies that used MHP focused on people above age 60 who tended to be already retired. For example, Ng and colleagues [25] examined MHP based on self-reported health indicators in community-dwelling people aged 60 to more than 85 years by using a latent class analysis. Quiñones and colleagues [26] analyzed MHP in an observation period of more than 16 years among people aged 65–95 years using indicators such as self-reported somatic diseases or functional status. Boehme and colleagues [24] advocated for including an indicator of mental health to better capture the health status of older people.

From these studies, we learned three things: First, among older people in general, there was considerable heterogeneity in terms of health if multiple dimensions were considered. Second, we learned that a variety of multi-morbidities can be observed. Third, morbidities themselves did not necessarily imply an impairment in terms of performing the activities of daily living.

However, the validity of these models has, thus far, not been tested for older workers. We therefore addressed this gap by examining the characteristics of MHP among workers aged 50–60 years at baseline. As noted, we have utilized the comprehensive model by Boehme and colleagues that included four indicators: chronic illness, functional limitation, self-perceived health, and depressive symptoms [24]. This led to our first research question (RQ1): What MHP do we find? In other words, how do older workers differ when these four dimensions of health are considered concurrently? Based on the literature, we hypothesized that some individuals would show a low level of health in one dimension but a high level in another.

- More precisely, we hypothesize to detect a MHP characterized by poor physical (i.e., chronic diseases and activity limitations) but good mental (i.e., self-perceived health and depressive symptoms) health (H1).

- Conversely, we also expect to discover a MHP characterized by good physical but poor mental health (H2).

## 3. Variability in health over the life course and by socioeconomic position

Lowsky and colleagues [27] pointed out that the relationship between chronological age and health status is much more variable than is often assumed. This implies that there is a large heterogeneity in health status within each age group, which is likely to increase over the life course. This heterogeneity has been explained in part by differences in terms of socioeconomic position (SEP). SEP is an important determinant of health and mortality for both younger and older people [28–30]. Moreover, according to the life course paradigm, health must be considered a dynamic phenomenon [31,32]. When these two bodies of literature are combined, they provide ample evidence that shows how health status and circumstances earlier in life (e.g., in childhood) affect health status later in life [33–36].

SEP is thus relevant for the life course dynamics of health. More specifically, the theory of cumulative disadvantages posits that differences in health increase over the life course [32,37–40]. However, it has been argued that life courses have become increasingly de-standardized, thereby leading to the belief that path-dependency based on SEP decreases [39]. Hence, the relevance of SEP as a determinant of health trajectory at the individual level may become less pronounced [41,42]. A methodological explanation for these disparate theories is provided by Fritzell and Rehnberg [28], who showed that growing or diminishing socioeconomic health inequalities over the life course are highly dependent on how we measure inequality, i.e., if we rely on absolute or relative measures [43].

Moreover, Wickrama and colleagues [20] pointed out that the use of analytical methods that are sensitive to intraindividual changes in health is still relatively rare. Most studies therefore may miss intraindividual differences in rates of change in health status. A study by Chang and colleagues [44] addressed this methodological shortcoming by examining how older people shifted between MHP over four measurement time points, with a focus on socioeconomic factors that explained these shifts. The study found that people with higher levels of education were more likely to experience stability in terms of their MHP, while people with lower levels of education were more likely to experience a decrease.

This led to our second research question (RQ2): How do the MHP vary by gender, age, and SEP (i.e., level of education and household income)?

- Based on the literature, we hypothesize that every MHP with a high share of poor health in any of the health dimensions is associated with low SEP (H3).

To gain a better understanding of how older workers move out of poor health statuses, we examined the stability of the MHP. More precisely, we asked whether older workers remained in their initial MHP within the observed time frame or if they moved out of it (RQ3). To address this research question, we examined the structural and individual stability of the MHP over a period of two years.

## 4. MHP and older workers' labor force participation

Many studies have found an association between poor health among older workers and their early exit to retirement [12,45]. However, this apparently clear association may mask distinct mechanisms in different groups of older workers [12]. In fact, a study that examined older workers' labor force participation above the statutory retirement age in 12 countries of Europe and the Americas, showed that the association between health and labor force participation was highly heterogeneous [46]. Using multichannel sequence analysis, the study found that both early retirement as well as late retirement can be associated with poor health. This heterogeneity has been explained, on the one hand, by poor health being associated with early exit from the labor force if workers' health conditions or their workplace situations did not allow them to continue working. This has been shown to be more frequently the case among manual workers [47]. On the other hand, poor health may be associated with, and potentially caused by, late retirement if workers are compelled to continue working for financial reasons [15].

This line of research led to our fourth research question (RQ4): How are the MHP associated with labor force participation? To address this question, we used employment status (e.g., employed or retired) and changes in working hours as the outcome measured six years after the baseline observation of the health status.

- We hypothesized that the MHP characterized by poor physical and good mental health is associated with a higher chance of being employed or a less strong reduction in working hours six years later (H4).

- Furthermore, we hypothesized that we will identify a MHP with low levels of all health indicators that will be associated either with a lower a chance of being employed or a higher reduction in working hours (H5).

- Finally, we hypothesized that we will identify a MHP with high levels of all health indicators that will be associated either with a higher chance of being employed or a lower reduction in working hours (H6).

## 5. Methods and data

### 5.1. Data and sample

We used longitudinal data of a period of six years from wave 5 (2013), wave 6 (2015), and wave 8 (2019) of the Survey of Health, Ageing and Retirement in Europe (SHARE) [48–51]. SHARE is a longitudinal observational panel survey that allows to track individuals over an extended period. For the study, we used wave 5 and 6 to identify the MPH and calculated the stability of the MPH between these two time points (w5 and w6). We then analyzed the relationship between the MPH from wave 6 with the outcome measure "current job situation", measured in wave 8, and with the outcome measure "change in working hours", measured as the difference between working hours in waves 6 and 8.

At the start of the observation period (wave 5), the sample comprised 13`279 older adults (mean age: 55.9 years; age Standard Deviation (SD) = 2.8 years; age range = 50–60 years; 54% female and 46% male). To be included in the sample, individuals had to be working—either employed or self-employed—and be aged 50–60 years during wave 5. In addition, in wave 8, information about the workers' current job situations had to be available. While in wave 5 individuals are all working, their job situation eight years later is more heterogeneous, reflecting the heterogeneity in retirement timing between and within European countries [52,53]. The sample selection criteria yielded an analytical sample of 5`006 individuals (mean age: 55.8 years; age SD = 2.79 years; age range = 50–60 years; 56% female and 44% male during wave 5). We included the countries for which data were available in waves 5, 6, and 8: Austria, Germany, Sweden, Spain, Italy, France, Denmark, Switzerland, Belgium, Israel, the Czech Republic, Luxembourg, Slovenia, and Estonia. The analyses of attrition rates showed that there were no differences between genders ($\chi2$ (1, N = 13`279) = 0.0361, p > 0.05). Different attrition rates were found between age groups, although the effect size (Cohen`s d) was small: the survey participation rate was higher among older versus younger workers (t(13,276) = −2.8.691, p < 0.001, d = .156; 95% confidence interval (CI) (0.120, 0.191)).

### 5.2. Ethics statement

The SHARE data collection procedures are continuously ethically reviewed. The activities of SHARE-ERIC related to human subjects' research are based on international research ethics principles such as the "Declaration of Helsinki". This specific study was reviewed and approved by the Ethics Council of the Max Planck Society in Munich. SHARE informs participants in advance using a formal invitation letter and a data protection statement in native language informing about the scientific purpose of the study and the voluntary participation. Informed consent is obtained verbally at the beginning of the survey-interview and is documented in the data. If the participant does not consent, this leads to the termination of the interview before the start. The SHARE data was accessed by the authors on 26.10.2022. The authors did not have access to information that could identify individual participants during or after data collection.

### 5.3. Measures

In accordance with the theoretical model developed by Boehme and colleagues [24], we used the following four categorical variables to identify the MHP in wave 5 (2013) and wave 6 (2015): (1) the number of chronic diseases (split into two

categories: fewer than two and two or more), (2) limitations in activities because of health (three categories: severely limited; limited, but not severely; not limited), (3) self-perceived health (two categories: very good/excellent; less than very good), and (4) depressive symptoms (two categories: yes and no). For the detailed operationalization of each variable see S1 File.

The measure "number of chronic diseases" summarizes the number of a choice of chronic diseases selected by survey participants from a list of chronic diseases. It thus relies on self-report and provides only a quantitative assessment of the number of chronic diseases; it does not consider the specific chronic diseases indicated. This measure is an indicator of people's health status in terms of multimorbidity. In Western societies, the prevalence of multimorbidity, i.e., the presence of multiple diseases in the same individual, is rising [54]. Research has shown that multimorbidity affects labor force participation. A study based on the SHARE dataset, for instance, has shown that chronic conditions together with high BMI negatively affected employment probability among older workers, and significantly increased the probability to retire early [55].

The measure "limitations in activities" describes how people experience health in everyday life [56]. It combines biological health (i.e., the bodily functions) and lived health (i.e., actual performance of activities) [57]. This measure serves as an important indicator of the extent to which people's health status affects their daily lives and can thus serve as a basis for the management of public health systems [56].

"Self-perceived (or, as it is in the literature often called, self-rated) health" is one the most common indicator in health research. It has in numerous studies proven to be a strong predictor of mortality (see, e.g., Lorem et al. [58]). It can be seen as a global health concept that incorporates both physical and psychological, or mental, states as evaluated in a cognitive process by the individual [59].

Finally, there exists a broad consensus in health research that an encompassing assessment of health also needs to include an indicator for mental health [60]. In the present study we include "depressive symptoms" in line with the model proposed by Boehme and colleagues [24]. Depressive symptoms have shown to be relevant with respect to labor force participation among older workers and are a predictor of early labor force exit [61,62].

With respect to their content, these variables may overlap. For instance, individuals with high levels of chronic diseases or depressive symptoms may also indicate low self-perceived health. The methodological approach used to created the MHP allows to – and even welcomes – the inclusion of overlapping health concepts. This approach allows to examine how the different health concepts coincide and does not lead to multicollinearity.

Since two of the variables – limitations in activities and depressive symptoms – are categorical variables, we transformed the other two variables – number of chronic diseases and self-perceived health – that have originally been interval-scaled into categorical variables.

Following Boehme et al., the selection of health indicators was based on content-related considerations [24]. Since some indicators—particularly *limitations in activities*—are categorical variables, categorical formats were also applied to the remaining indicators for consistency reasons. These categorical variables were created by the SHARE data team and have already been used in this form in numerous studies.

We used two outcome measures. First, we included the workers' current job situations at wave 8 with the following categories: retired, employed or self-employed, unemployed, permanently sick or disabled, homemaker, and other. Second, we calculated the changes in working hours by subtracting the working hours in wave 8 from those in wave 6.

The indicators of socio-demography and socioeconomic position that we included were age, gender, level of education, and relative household income (measured as quintiles within each country) at wave 5 and 6. The dataset we used relied on a binary categorization of gender (male/female). Participants' gender was identified by the interviewer with the option to directly ask the participant [63].

## 5.4. Statistical analyses

We used longitudinal data over six years from waves 5, 6, and 8 of SHARE and conducted a latent transition analysis (LTA), an extension of latent class/profile analysis across multiple measurement time points [64–66]. LTA is particularly

well suited for identifying patterns within distinct subgroups – so-called classes – over time and for assessing both structural stability of identified patterns of the classes (same shape of the patterns over time) and individual stability (probability that an individual remains in the same class over time). For an in-depth discussion and comparison of LTA with other analysis see for example Morin and Litalien or Oberski [23,67].

LTA requires a stepwise approach, whereby cross-sectional latent class analysis as well as the selection of the number of patterns were conducted separately for each measurement time point. In a second step, the two measurement time points were connected in a longitudinal model of the latent transition analysis to assess the stability of the patterns over time. To identify the latent MHP over time, we used waves 5 and 6 (RQ1) and conducted separate cross-sectional latent class analyses with each of the two measurement time points (waves 5 and 6) with the following four health indicators: number of chronic diseases, limitations in activities, self-perceived health, and depressive symptoms. Latent class analysis was chosen over latent profile analysis because it is suitable for categorical variables.

In the initial step, we separately computed two to six MHP for each measurement time point. We then followed a stepwise modeling strategy to identify the best-fitting number of latent classes (MHP). We began by estimating models with an increasing number of classes separately for each measurement time point (waves 5 and 6). To determine the optimal number of MHP, we considered both statistical and content-related criteria.

For the statistical evaluation, we employed Log-likelihood (LL), the Bayesian information criterion (BIC), sample-adjusted BIC, the bootstrap likelihood ratio test (BLRT), and the Vuong-Lo-Mendell-Rubin likelihood ratio test (VLMR). We applied the elbow criterion for the BIC and the sample-adjusted BIC. In terms of content-related criteria, we considered the principles of parsimony, conformity with theoretical considerations, and the interpretability of the results.

Next, to gain a better understanding of the characteristics of the individuals in each MHP, we performed Wald tests with the following variables: age, gender, level of education, and relative household income (RQ2). Wald tests compare the equality of means or distributions across classes [68]. Finally, to assess the structural stability of the MHP over time (RQ3a), we calculated the average squared Euclidean distance between the most similar class MHP at waves 5 and 6 [69]. Additionally, to explore the individual stability of the MHP over time (RQ3b), we calculated transition probabilities [65]. Transition probabilities indicate the likelihood that an individual will move between or remain in certain patterns over time, serving as a measure of individual stability.

To investigate the associations between the MHP and working hours as well as the association between MHP and employment status (RQ4), we used Wald tests [68,70]. Finally, we assessed whether class membership varied systematically by country, cross-tabulating the four MHP with the 15 countries in our sample, reporting the $\chi^2$ statistic and Cramer's V (see S2 Table). Missing data for all four health indicators are below 5% (see Table 2). To handle the missing data, we employed the full information maximum likelihood procedures (FIML) [71].

We performed all analyses in Mplus Version 8.8 [72], using maximum likelihood estimation (MLR) with robust standard errors.

## 6. Results

### 6.1. MHP

To identify the latent MHP, we tested different models ranging from two to six MHP for both wave 5 and wave 6. Overall, we observed improvements in LL, BIC, and sample-adjusted BIC as we increased the number of MHP (see Table 1). The elbow criterion suggested an optimum in the range of three to five MHP. Additionally, the BLRTs yielded significant results for all of the models, while the VLMR pointed to an optimum of a four-MHP model.

We chose the four-class solution after considering both statistical outcomes and substantive interpretability. While statistical indicators suggested an optimal range between three and five classes, the four-class solution was selected because it provided the clearest conceptual differentiation between health patterns that aligned closely with previous literature and our hypotheses. The four-MHP model thus was the most suitable model for both waves.

**Table 1. Fit indices of the MHP over time.**

| Classes wave 5 | npar | LL | AIC | SABIC | BIC | VLMR | BLRT | c1 | c2 | c3 | c4 | c5 | c6 |
|---|---|---|---|---|---|---|---|---|---|---|---|---|---|
| 1 | 5 | −12238 | 24485 | 24502 | 24518 | | | 5006 | | | | | |
| 2 | 11 | −11391 | 22803 | 22840 | 22875 | 0.000 | 0.000 | 1610 | 3397 | | | | |
| 3 | 17 | −11367 | 22768 | 22825 | 22879 | 0.000 | 0.000 | 666 | 1977 | 2364 | | | |
| 4 | 23 | −11361 | 22769 | 22845 | 22919 | 1.000 | 0.026 | 427 | 2878 | 1131 | 571 | | |
| 5* | 29 | −11358 | 22774 | 22871 | 22963 | 0.456 | 0.108 | 818 | 363 | 39 | 3041 | 746 | |
| 6* | 35 | –11358 | 22786 | 22902 | 23014 | 0.630 | 0.773 | 604 | 2679 | 533 | 369 | 208 | 614 |
| Classes wave 6 | npar | LL | AIC | SABIC | BIC | VLMR | BLRT | c1 | c2 | c3 | c4 | c5 | c6 |
| 1 | 5 | −12623 | 25256 | 25273 | 25288 | | | 5006 | | | | | |
| 2 | 11 | −11754 | 23530 | 23567 | 23602 | 0.000 | 0.000 | 1813 | 3193 | | | | |
| 3 | 17 | −11727 | 23487 | 23544 | 23598 | 0.208 | 0.000 | 391 | 1672 | 2943 | | | |
| 4 | 23 | −11717 | 23479 | 23556 | 23629 | 0.002 | 0.004 | 1268 | 259 | 536 | 2943 | | |
| 5* | 29 | −11716 | 23490 | 23587 | 23679 | 0.373 | 0.511 | 948 | 227 | 504 | 2782 | 545 | |
| 6* | 35 | –11716 | 23502 | 23619 | 23730 | 0.055 | 1.000 | 619 | 356 | 432 | 1665 | 1906 | 28 |

npar = number of parameters; LL = Log-likelihood; AIC = Akaike information criterion; BIC = Bayesian information criterion; SBIC = sample-adjusted Bayesian information criterion; VLMR = Vuong-Lo-Mendell-Rubin likelihood ratio test; BLRT = bootstrap likelihood ratio test; c1 = size of class 1; c2 = size of class 2; c3 = size of class 3; c4 = size of class 4; c5 = size of class 5; c6 = size of class 6. *Estimation problems.

We identified four MHP (see Table 2) that we named as follows: (1) overall healthy workers, (2) workers with moderate limitations and low self-perceived health, (3) physically healthy workers with low self-perceived health and depressive symptoms, and (4) workers in overall poor health. None of the MHP differed significantly by age (p > 0.05). The MHP-by-country table was significant ($\chi^2 = 254.32$, df = 39, p < 0.001), yet Cramer's V = 0.13, denoting a small association. All 15 countries contributed cases to every MHP; the largest single-country share in any class was 21.3% (Czech Republic for cluster 4; see S2 Table).

The first MHP, [1] overall healthy workers, included the largest number of observations (wave 5: N = 2`878, 58%; wave 6: N = 2`943, 59%). Individuals in this MHP were characterized by high levels in all health indicators: they had fewer than two chronic diseases and were not limited in their activities (neither severely nor moderately). Only a third of them considered their health as less than very good. In addition, individuals had no depressive symptoms. In this MHP, we found an equal gender distribution. With 50% of the individuals having tertiary education, this was the MHP with the highest level of education as well as an above average household income (23% in the highest quintile) compared to the other MHP. The latter finding was in line with the literature on the socioeconomic inequalities within health that emphasize the strong association between SEP and health outcomes at a given time point (especially in older age) as well as on the life-course dynamics of health.

In MHP 2: workers with moderate limitations and low self-perceived health (wave 5: N = 1`131, 23%; wave 6: N = 1`268, 25%), half of the individuals had more than two chronic diseases. Most of the individuals in this second MHP indicated moderate but not severe limitations in activities. They did perceive their health as less than very good but did not indicate having any depressive symptoms. This second MHP was characterized by 69% males, an average level of education (41% post-secondary, non-tertiary education), and an average household income.

In MHP 3: physically healthy workers with low self-perceived health and depressive symptoms (wave 5: N = 427, 8%; wave 6: N = 259, 5%), the individuals did not have more than two chronic diseases and did not indicate any limitations in activities. However, this MHP was characterized by a low level of self-perceived health and a very high share of people

**Table 2. Characteristics of the MHP and their associated outcomes.**

| | | | Overall sample | Overall healthy workers | Workers with moderate limitations and low self-perceived health | Physically healthy workers with low self-perceived health and depressive symptoms | Workers in overall poor health |
|---|---|---|---|---|---|---|---|
| **Wave 5** | Missing data | MHP size | 100%; N=5`007 | 58%; N=2`878 | 23%; N=1`131 | 9%; N=427 | 11%; N=571 |
| **Number of chronic conditions** | N=4 (0.1%) | Fewer than 2 diseases | 73% | 91% | 55% | 88% | 24% |
| | | 2+chronic diseases | 27% | 9% | 45% | 12% | 76% |
| **Limitations with activities** | N=0 (0.0%) | Severely limited | 5% | 1% | 4% | 2% | 26% |
| | | Limited, but not severely | 21% | 3% | 49% | 13% | 51% |
| | | Not limited | 74% | 97% | 48% | 85% | 23% |
| **Self-perceived health** | N=1 (<0.0001%) | Very good/excellent | 44% | 67% | 17% | 51% | 0% |
| | | Less than very good | 56% | 33% | 84% | 49% | 100% |
| **Depressive symptoms** | N=62 (1.2%) | No | 81% | 100% | 88% | 0% | 41% |
| | | Yes | 19% | 0% | 12% | 100% | 59% |
| **Wave 6** | | MHP size | | 59%; N=2`943 | 25%; N=1`268 | 5%; N=259 | 10%; N=536 |
| **Number of chronic conditions** | N=1 (<0.0001%) | Fewer than 2 diseases | 41% | 92% | 50% | 80% | 30% |
| | | 2+chronic diseases | 59% | 8% | 50% | 21% | 70% |
| **Limitations with activities** | N=1 (<0.0001%) | Severely limited | 6% | 1% | 8% | 1% | 30% |
| | | Limited, but not severely | 26% | 5% | 52% | 28% | 55% |
| | | Not limited | 68% | 94% | 41% | 70% | 15% |
| **Self-perceived health** | N=3 (<0.0001%) | Very good/excellent | 41% | 67% | 14% | 27% | 1% |
| | | Less than very good | 59% | 33% | 87% | 73% | 99% |
| **Depressive symptoms** | N=74 (1.5%) | No | 81% | 95% | 100% | 11% | 28% |
| | | Yes | 19% | 5% | 0% | 89% | 72% |
| **Structural stability** | | Average squared Euclidean distance | | <0.005 | 0.04 | 0.25 | 0.08 |
| *Descriptive Variables* | | | | | | | |
| **Gender w5 Overall test X²=48.46, df=3, p<0.001** | N=0 (0.0%) | Male | 44% | 51% [3] | 69% [3, 4] | 21% [1, 2] | 31% [2] |
| | | Female | 56% | 49% | 31% | 79% | 69% |
| **Age w5** | N=0 (0.0%) | Mean overall test chi-square=31.189, p<0.001 | 55.80 | 55.6 [2, 3] | 56.4 [1, 3, 4] | 55.2 [1, 2] | 55.7 [2] |
| **Household income (quintile) w5 Overall test X²=71.06, df=12, p<0.001** | N=0 (0.0%) | 1-20% | 20% | 18% | 21% | 23% | 27% |
| | | 20.1-40% | 20% | 18% | 22% | 21% | 23% |
| | | 40.1-60% | 20% | 21% | 19% | 22% | 19% |
| | | 60.1-80% | 20% | 21% | 20% | 17% | 19% |
| | | 80.1-100% | 20% | 23% | 18% | 18% | 13% |
| **Education w5 Overall test chi-square=1138.72, df=9, p<0.001** | N=0 (0.0%) | Primary and secondary education | 17% | 13% [2, 3, 4] | 19% [1, 3] | 0% [1, 2, 4] | 21% [1, 3] |
| | | Post-secondary, non-tertiary education | 47% | 42% | 50% | 46% | 51% |
| | | Tertiary education | 35% | 45% | 29% | 50% | 27% |
| | | Other | 2% | 1% | 2% | 3% | 2% |

*(Continued)*

| | | | Overall sample | Overall healthy workers | Workers with moderate limitations and low self-perceived health | Physically healthy workers with low self-perceived health and depressive symptoms | Workers in overall poor health |
|---|---|---|---|---|---|---|---|
| *Outcomes* | | | | | | | |
| **Current job situation w8**<br>**Overall test**<br>**X² = 166.49, df = 15, p < 0.001** | N = 0 (0.0%) | Retired | 36.4% | 32% [2, 3, 4] | 51% [2, 3, 4] | 37% [1, 2, 4] | 33% [1, 2, 3] |
| | | Employed or self-employed | 55.9% | 64% | 45% | 52% | 44% |
| | | Unemployed | 2.5% | 1% | 0% | 5% | 5% |
| | | Permanently sick or disabled | 2.4% | 1% | 1% | 3% | 15% |
| | | Homemaker | 1.4% | 1% | 0% | 3% | 2% |
| | | Other | 1.4% | 1% | 3% | 1% | 1% |
| **Change in working hours (w8-w6)** | N = 745 (14.9%) | Overall test chi-square = 19.216, p < 0.001 | −12.33 h | −10.8 h [2] | −14.9 h [1] | −14.5 h | −12.4 h |

Significant differences between the MHP are shown in square brackets. For the household income, we used the values from the chi-square test on a "manifest level" due to convergence problems with the auxiliary variable.

having felt sad or depressed in the month before the interview. This third MHP was characterized by the highest share of females (79%) compared to the other MHP. It had the highest percentage of individuals with the lowest level of education (28% with primary and secondary education) and the lowest household income (23% of individuals in the lowest quintile). This was the MHP with the smallest sample size.

MHP 4: workers in overall poor health (wave 5: N = 571, 11%; wave 6: N = 536, 10%), was characterized by a comparably high share of individuals with multiple chronic diseases (76%) and with severe (26%) or moderate (51%) limitations in activities. Furthermore, all of the individuals in this MHP perceived their health as less than very good, and 59% of them showed depressive symptoms. Workers in this MHP were predominantly female (70%) and had an average level of education (42% post-secondary, non-tertiary education and 42% tertiary education), with a less-than-average household income (27% were in the lowest quintile).

## 6.2. Structural and individual stability of the MHP

All four MHP showed high structural stability. This followed from the finding that the average squared Euclidean distance between the most similar class MHP at waves 5 and 6 was between < 0.005 and 0.25 (see Table 2). Fig 1 illustrates this by the similar shapes of the MHP in wave 5 (on the left-hand side) and wave 6 (on the right-hand side). Moreover, Fig 1 makes clear the individual stability of the MHP. In other words, it illustrates the transition probabilities that indicated individuals' probability to move or stay in the same MHP between waves 5 and 6. We found that the large majority of the older workers remained in the same MHP (83% to 93% transition probabilities). However, one exception was MHP 3: physically healthy workers with low self-perceived health and depressive symptoms. Individuals in this MHP had a probability of only 41% of remaining in the same MHP, while 56% transitioned to MHP 1: overall healthy workers.

## 6.3. MHP and their associations with current job situation and working hours

The four MHP significantly differ in the current job situations at wave 8 (p < 0.001) (Table 2). Additionally, significant differences were observed between the four MHP concerning changes in working hours between wave 6 and wave 8 (p < 0.001). All workers included in our sample were employed or self-employed in wave 5. Six years later, more than half of the MHP

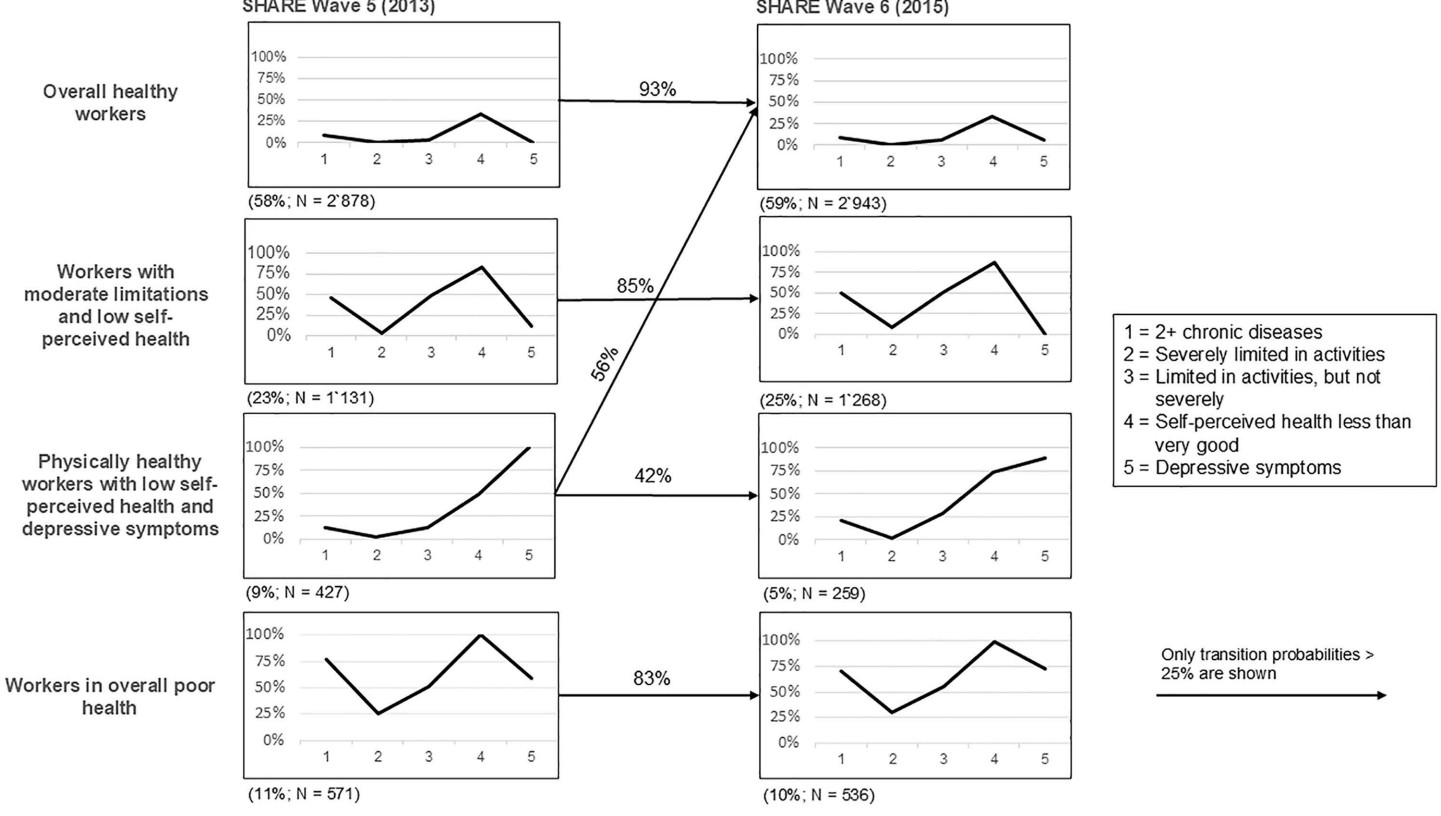

**Fig 1. Latent multidimensional MHP over time and transition probabilities.** The x-axis of Fig 1 indicates only the lower levels of health, e.g., we show only the share of individuals indicating two or more chronic diseases, while the share of individuals indicating fewer than two chronic diseases is not explicitly shown.

1 overall healthy workers (64%) and the MHP 3 physically healthy workers with low self-perceived health and depressive symptoms (52%) were still working in a job. In contrast, 51% of the MHP 2 workers with moderate limitations and low self-perceived health were retired six years after baseline. Furthermore, the MHP 4 workers in overall poor health had the comparably highest rate of unemployment (5%) and the highest share of permanently sick or disabled individuals (15%).

We found a significant difference between MHP 1 and 2. While the MHP 2 workers with moderate limitations and low self-perceived health had reduced their working hours by an average of 14.9 hours, the MHP 1 overall healthy workers had reduced their working hours by an average of only 10.8 hours. The MHP 3 physically healthy workers with low self-perceived health and depressive symptoms had experienced a reduction of working hours similar to those in MHP 2 (a reduction of 14.5 hours); however, the differences between MHP 3 and the other MHP were not statistically significant. The MHP 4 workers with overall poor health were second least likely to decrease their working hours (a reduction of 12.4 hours on average). However, the differences between MHP 4 and the other MHP were not statistically significant.

## 7. Discussion

In this study, we examined the relationship between older workers' health and their labor force participation. Using waves 5, 6, and 8 of the Survey of Health, Ageing and Retirement in Europe (SHARE), we began by identifying MHP, examined the stability of these MHP after two years, and finally analyzed how the MHP were related to employment status and working hours six years later.

First, we identified four distinct MHP among workers aged 50–60. While many workers experienced good health in all dimensions (MHP 1) or poor health in all dimensions (MHP 4), some individuals experienced a high level of health in one dimension while experiencing a low level in another dimension (MHP 2 and 3). Identifying MHP 2: workers with moderate limitations and low self-perceived health supported our hypothesis H1, which expected that we would find a MHP with poor physical but good mental health. Identifying MHP 3: physically healthy workers with low self-perceived health and depressive symptoms supported our hypothesis H2, which expected that we would discover a MHP with good physical but poor mental health. MHP 2 was much more frequent (23% of the sample) than MHP 3 (9%). These findings highlight the importance of examining health as a multidimensional construct [20].

Second, we were not able to confirm hypothesis H3, which stated that every MHP with high shares of poor health in one of the health dimensions would be associated with low SEP. In contrast, only MHP 4, with poor health in all of the health dimensions, was clearly associated with low SEP. MHP 2 and 3 consisted of a similar share (43%) of individuals with low household income, but MHP 3 had a substantially higher share of people with high levels of education (97% of people with more than secondary education) than MHP 2 (79%). From the perspective of socioeconomic differences in health, this result adds new evidence to the existing literature that indicates that employment is beneficial for individuals with depressive symptoms [73,74]. While the literature generally describes a direct link between poor health and low SEP [38], the results in our study differentiated this association by showing that physical and mental dimensions of health are not always aligned [75]. In fact, none of our multidimensional MHP identified a clear link between low SEP and poor *mental* health. This link was only evident for *physical* health measures.

Third, we found that most of the MHP remained stable over time. However, approximately half of the individuals in MHP 3: physically healthy workers with low self-perceived health and depressive symptoms, shifted to MHP 1: overall healthy workers. One explanation for this result may be that there is relatively high potential for recovery from depressive symptoms indicating that they are more temporary and changeable health conditions [76]. Another explanation of this result may be that we only captured individuals with mild depressive symptoms. This explanation is supported by research on the validity of a single-question screening to detect depressive symptoms. While such instruments generally perform well at capturing people with objectively present depressive symptoms, this indicator also carries the risk of false positives, i.e., people who report having felt depressed but who are not clinically depressed [77]. Furthermore, it should be noted that our analytical design, in which all of the individuals were working at baseline, most likely resulted in us not including people with the most severe and especially persistent depressive symptoms over a longer period of the life course [78].

Fourth, we found significant differences in labor force participation between older workers in the four MHP. Workers in MHP 1: overall healthy workers showed, as expected, the highest levels of employment or self-employment six years after the baseline assessment. They also were the least likely to have reduced their working hours. This provides clear support for our hypothesis H6, which expected that a MHP with high levels of all health indicators would be associated with a higher chance of being employed or a lower reduction in working hours. In contrast, workers in MHP 4: overall poor health had the comparably lowest labor force attachment. This provides clear support for our hypothesis H5 that expected a MHP with low levels of all health indicators which is associated with a lower a chance of being employed or a higher reduction in working hours. On the other hand, hypothesis H4 anticipated that we would identify a MHP with poor physical and good mental health that would be associated with stronger attachment to the labor force. We did not identify such a MHP; accordingly, H4 is falsified. This result may be explained by a higher likelihood of workers with poor physical health to leave the labor force due to disability. Previous research indeed shows that workers with poor physical health – often lower educated manual workers – are more likely to qualify for disability benefits than workers with poor mental health ([75]).

However, we identified MHP 3: physically healthy workers with low self-perceived health and depressive symptoms which had a comparatively high labor force attachment. As noted, this may be related to employment being able to act as a buffer and alleviate depressive symptoms ([73]), but can also be explained by the fact that people with poor mental health remain in the labor force but are on long-term sick leave. As the variable "depressive symptoms" is assessed in

the SHARE data referring to the experience of the past month, it is furthermore possible that it reflects only a temporary condition. This underscores the usefulness of using a multidimensional health construct in examining the impact of health on labor market participation as it allows to combine variables assessing a temporary condition with variables assessing more permanent conditions such as self-rated health. Overall, our finding highlights the importance of using a multidimensional construct to examine labor force participation: a low level of health in one dimension does not necessarily mean that labor force participation is threatened. Admittedly, our multidimensionality is still somewhat crude and could be further elaborated in future research. Nevertheless, our approach provides a more comprehensive understanding than previously available for why some older people with health impairments remain active in the workforce and others do not [20].

These findings contribute to the literature that has demonstrated that despite high labor force exit rates among older workers with poor health, poor health *per se* is not the only reason for quitting the labor force [12]. Rather, labor force participation appears to depend on the specific aspects of workers' health that are impaired in addition to the working conditions (e.g., the possibility of adjusting the workplace,; see [17] and institutional factors (e.g., the amount of the pension benefit; see [46]. Accordingly, some health problems are likely to be more detrimental than others with respect to work and may serve as a turning point in the employment trajectories of workers [79]. Our results suggest that experiencing poor physical health—particularly multiple chronic conditions—is much more likely to present such a turning point and thus creates discontinuity in the life course of older workers.

## 7.1. Implications, limitations, and future research directions

Older workers face many obstacles with respect to labor market reintegration [80], which are likely exacerbated for older workers with health problems. It has been shown that active labor market policies (ALMP) may improve labor market prospects for older workers, particularly when programs are designed to address the specific needs of this group [81]. If programs are designed to also address health problems among older workers, such programs could potentially increase labor market participation. In a randomized controlled trial study from Germany, it was shown that a small group-intervention among older health care workers at the workplace lead to an improvement of their mental health-related well-being [82]. Our findings suggest that ALMP may be particularly beneficial for older workers with depressive symptoms and moderate activity limitations and may improve their chances of remaining in the labor force.

A first limitation of our study is that it did not allow for causal inference. Although our sample included only people who were working at the time of the baseline assessment, it is possible that we encountered a healthy worker effect (i.e., that our sample included only the healthiest share of workers because those with really poor health had already quit the labor force at an earlier age) [83]. Additional confounding factors that were not included in our study may need to be considered.

A second limitation is that we find age differences in attrition rates with lower attrition rates among older workers. This may lead to an overestimation of individuals who are in retirement because they reached statutory retirement age and thus of an underestimation of the effect of poor health on labor force participation.

A third limitation is the lack of consideration of macro-level aspects, such as welfare and labor market programs [84], which could have an impact on the participation of people with long-term illnesses [85]. Given the large number of European countries included in the SHARE dataset, there is potential for consideration of these aspects. However, examining differences between countries or welfare regimes would have been beyond the scope of the present article. Such an approach may, however, offer directions for future research.

Finally, since we pooled data from 15 countries, national nuances may be attenuated. Country explains only a small share of variance in class membership (Cramer's V = 0.13), although one country contributes 21.3% to a single class. Future work should apply multilevel or country-specific latent-class approaches to explore macro-level influences more fully.

As discussed earlier, one potential arena for future research would be to analyze whether different strategies in terms of ALMP are associated with different outcomes. Typically, ALMP are analyzed as an aggregate category of government

spending [84], but they come in different forms. For our purposes, there are, for example, more detailed indicators of ALMP that are targeted toward the disabled or those with long-term illnesses, which would permit a more nuanced analysis of the types of policies that different welfare states are implementing and whether these are associated with increased chances of labor market retention [86,87].

**Declaration of generative AI and AI-assisted technologies in the writing process:** During the preparation of this work the author(s) used DeepL.com and Grammarly.com in order to improve readability and language of the manuscript. After using this tool/service, the authors reviewed and edited the content as needed and take full responsibility for the content of the publication.

## Supporting information

**S1 File. Operationalization of the used health indicators.**
(PDF)

**S2 Table. Distribution of latent classes by country (row %).**
(PDF)

## Author contributions

**Conceptualization:** Isabel Baumann, Vanessa Gut, Rainer Gabriel, Daniel Fredriksson, Johan Fritzell.

**Data curation:** Vanessa Gut.

**Formal analysis:** Vanessa Gut.

**Funding acquisition:** Isabel Baumann.

**Methodology:** Vanessa Gut.

**Project administration:** Isabel Baumann.

**Visualization:** Vanessa Gut.

**Writing – original draft:** Isabel Baumann, Vanessa Gut, Rainer Gabriel, Daniel Fredriksson, Johan Fritzell.

**Writing – review & editing:** Isabel Baumann, Vanessa Gut, Rainer Gabriel, Daniel Fredriksson, Johan Fritzell.

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
