## [Decision Letter · Decision Letter 0]

8 Jul 2025

PONE-D-25-21578Multidimensional health patterns and labor market participation among older workers:

Evidence from a European six-year follow-up studyPLOS ONE

Dear Dr. Baumann,

Thank you for submitting your manuscript to PLOS ONE. After careful consideration, we feel that it has merit but does not fully meet PLOS ONE’s publication criteria as it currently stands. Therefore, we invite you to submit a revised version of the manuscript that addresses the points raised during the review process.

I have received two reports from respected reviewers. Reviewer #2 suggests a minor revision, while Reviewer #1 recommends a major revision. Overall, I agree with the comments provided by both reviewers, and I believe it is necessary to address their suggestions thoroughly.

Therefore, I have classified your submission as requiring a major revision and invite you to revise and resubmit your manuscript accordingly.

Please ensure that your revised submission includes a detailed response to each reviewer comment (particularly, for Reviwer#1).

We look forward to receiving your revised manuscript.

Kind regards,

Edgar Demetrio Tovar-García

Academic Editor

PLOS ONE

Journal Requirements:

[This work was supported by the Swiss National Science Foundation (SNSF) (grant number 179696: IB, VG; https://data.snf.ch/grants/grant/179696); the ZHAW Schwerpunkt Angewandte Gerontologie [Research focus “Applied Gerontology”: RG; https://www.zhaw.ch/de/forschung/departementsuebergreifende-kooperationen/angewandte-gerontologie]; Forte (grant numbers 2016-07206: JF; and 2021-00999: DF; https://forte.se/en/about-forte/our-mission/). The funders had no role in study design, data collection and analysis, decision to publish, or preparation of the manuscript.].

Reviewers' comments:

Reviewer's Responses to Questions

**Comments to the Author**

1. Is the manuscript technically sound, and do the data support the conclusions?

Reviewer #1: Partly

Reviewer #2: Yes

2. Has the statistical analysis been performed appropriately and rigorously? 

Reviewer #1: I Don't Know

Reviewer #2: Yes

3. Have the authors made all data underlying the findings in their manuscript fully available?

Reviewer #1: Yes

Reviewer #2: Yes

4. Is the manuscript presented in an intelligible fashion and written in standard English?

Reviewer #1: Yes

Reviewer #2: Yes

5. Review Comments to the Author

Reviewer #1: The authors address an interesting and socially relevant topic. They argue that health is a multidimensional construct and advocate for using a multidimensional health measure to examine the role of health in labor market participation at older working ages. Their approach represents a small but significant contribution to the field of health, employment and retirement research.

The authors analyzed data from 5,006 older workers participating in the Survey of Health, Ageing and Retirement in Europe (SHARE), they performed a latent transition analysis based on four health indicators - chronic conditions, activity limitations, self-rated health, and depressive symptoms - to identify multidimensional health profiles (MHPs). In the second step, the examined the relationship between these MHPs and labor force participation six years later. Four distinct patterns were identified. The authors conclude that using health as a multidimensional construct allows to better understand the social implications of poor health.

In addition to the research idea, I also appreciate the choice of dataset for this investigation. However, there are several major—primarily conceptual—issues with the paper that, in my view, must be addressed before it can be considered for publication.

Methodology

The manuscript would gain from more clarity in the choice of the methodological approach and its description.

- Multidimensional construct: What is the author’s reasoning behind the indicators chosen for the mc? The fact that Boehme has chosen this? Why is this model superior to others?

- Why using overlapping health concepts in the model?

- Why using dichotomous health indicators if linear indicators can also be used in latent group modelling?

- From my point of view the methodology latent transition analysis needs to be explained more in detail. How does one have to understand the clusters if the numbers of participants in each cluster differ by wave? How do you estimate belonging to a cluster? On which basis (wave 5 or 6? or a transition measure) did the authors calculate the distribution of the outcomes?

- What does it mean for analyses and findings to use an initial sample with an age range from 50 to 60 years and following these for eight years, with the older reaching/passing statutory retirement age and the younger still being far away from it?

Measurement of health

If concepts of health are in the focus of the paper, a detailed description, presentation and discussion of each health indicator chosen should be given.

- What is known about each measure? What are the underlying health concepts of each measure used? What is known about stability over time, predictivity, validity? How were the measures operationalized – and why? Distributions in the sample.

- For example: is the term “depressive symptoms” justified if the underlying question is whether the respondent has “felt sad or depressed in the month before the interview”. Doesn’t the high proportion of participants shifting from Cluster 3 (wave 5) to Cluster 1 (wave 6) indicate that the question reflects a temporary condition, possibly linked to a specific event or living situation?

- The understanding of “self-perceived health” seems too limited in the manuscript. Why should it be an indicator for mental health? (p5) Several authors have reasoned about what this highly predictive indicator stands for. I recommend reading Jylhä 2009 “What is self-rated health and why does it predict mortality? Towards a unified conceptual model”.

- Number of chronic diseases: 1 vs 2+. Why this dichotomization? Which disease groups are included?

Further measures

- Measuring household income: Why not use "household net equivalized income," which more accurately reflects social status than total household income alone? This should be possible with SHARE data.

- Measuring change of weekly working hours. The finding is very surprising that on average, the workers have reduced their weekly working hours by >> 10 hours over 6 years. Which participants were included in the calculation: also those that had stopped working for full? In that case the overall mean is not useful for analysis. Or: was it due to some specific countries with special regulations allowing for such a substantial reduction of working time prior to retirement? This however, would raise the question of heterogeneity between countries.

Cultural heterogeneity

- One important factor that may explain many of the findings in the manuscript is the strong country-level influence on the topics of (work,) health, employment, and retirement. These areas are highly context-dependent, shaped by national welfare systems, retirement regulations, social norms, and cultural perceptions of health. From my perspective, it is crucial for a study of this kind to discuss the potential implications of analyzing pooled data from 14 European countries as well as Israel. What does this heterogeneity mean for the findings? How are participants from different countries distributed across the identified clusters? Are there clusters predominantly composed of individuals from specific countries and vice versa? Were any sensitivity analyses conducted based on welfare state regimes or country groupings?

Overall

The authors acknowledge that their concept of "multidimensionality is still somewhat crude." I appreciate this critical reflection and would encourage the authors to reconsider their methodological approach. From my point of view not much is gained by specifically applying this approach, however, there are more differentiated ways of clustering health conditions in populations. Overall: did the approach applied really contribute to the central aim stated in the introduction: “These facts raise our principal question of what factors influence whether older workers with adverse health conditions exit or remain in the labor force.”?

Reviewer #2: This manuscript addresses an important and timely topic: the identification of Multidimensional Health Profiles (MHP) among older workers and their association with employment status and working time reduction. The study uses high-quality longitudinal data from the SHARE survey and applies Latent Transition Analysis (LTA), which is an appropriate and advanced method to capture temporal health profile dynamics.

The findings offer valuable insights into the complex interplay between health and employment in later life, with clear implications for policy and active ageing strategies. The manuscript is clearly written and logically structured.

However, there are some areas that could be improved to enhance clarity and impact.

-Conceptual clarity

The definition of “Multidimensional Health Profile (MHP)” should be introduced more explicitly earlier in the introduction, before stating the hypotheses.

The formulation of hypotheses could be streamlined for better readability.

-Methodology

Although LTA is well applied, the rationale for choosing this method over alternatives should be made clearer.

The manuscript could benefit from a more critical discussion of the SHARE sample design, particularly regarding rotation and attrition across waves.

-Discussion and implications

The discussion could be strengthened by elaborating on the practical implications of the identified health profiles for policies on active ageing, employment support, and reducing inequalities.

It would also be valuable to comment on the potential cross-national variability and whether institutional differences may limit generalizability.

-Limitations

A more detailed reflection on sample attrition and its potential influence on MHP stability would be beneficial.

The authors could acknowledge the reliance on self-reported health measures as a possible limitation.

Recommendation: Accept with minor revisions

This is a strong and relevant contribution to research on health and employment among older adults. The suggestions above are meant to improve clarity and robustness, but they do not challenge the study's overall validity. With minor revisions, the manuscript will be ready for publication.

6. PLOS authors have the option to publish the peer review history of their article (what does this mean? ). If published, this will include your full peer review and any attached files.

**Do you want your identity to be public for this peer review?** For information about this choice, including consent withdrawal, please see our Privacy Policy .

Reviewer #1: No

Reviewer #2: No

---

## [Author Response · Author response to Decision Letter 1]

24 Aug 2025

We thank the Editor and the two Reviewers for the valuable comments. Please find our point-by-point responses in the RESPONSE TO REVIEWERS.

---

## [Decision Letter · Decision Letter 1]

17 Sep 2025

Multidimensional health patterns and labor market participation among older workers:

Evidence from a European six-year follow-up study

PONE-D-25-21578R1

Dear Dr. Baumann,

We’re pleased to inform you that your manuscript has been judged scientifically suitable for publication and will be formally accepted for publication once it meets all outstanding technical requirements.

Kind regards,

Edgar Demetrio Tovar-García

Academic Editor

PLOS ONE

Additional Editor Comments (optional):

Reviewer #1:

Reviewers' comments:

Reviewer's Responses to Questions

**Comments to the Author**

1. If the authors have adequately addressed your comments raised in a previous round of review and you feel that this manuscript is now acceptable for publication, you may indicate that here to bypass the “Comments to the Author” section, enter your conflict of interest statement in the “Confidential to Editor” section, and submit your "Accept" recommendation.

Reviewer #1: All comments have been addressed

2. Is the manuscript technically sound, and do the data support the conclusions?

Reviewer #1: Yes

3. Has the statistical analysis been performed appropriately and rigorously? 

Reviewer #1: Yes

4. Have the authors made all data underlying the findings in their manuscript fully available?

Reviewer #1: Yes

5. Is the manuscript presented in an intelligible fashion and written in standard English?

Reviewer #1: Yes

6. Review Comments to the Author

Reviewer #1: (No Response)

7. PLOS authors have the option to publish the peer review history of their article (what does this mean? ). If published, this will include your full peer review and any attached files.

**Do you want your identity to be public for this peer review?** For information about this choice, including consent withdrawal, please see our Privacy Policy .

Reviewer #1: No

---

## [Editor Report · Acceptance letter]

PONE-D-25-21578R1

PLOS ONE

Dear Dr. Baumann,

I'm pleased to inform you that your manuscript has been deemed suitable for publication in PLOS ONE. Congratulations! Your manuscript is now being handed over to our production team.

Kind regards,

on behalf of

Dr. Edgar Demetrio Tovar-García

Academic Editor

PLOS ONE